# Supervised Contrastive Learning

**Prannay Khosla** [*]
Google Research

**Piotr Teterwak** [*][†]
Boston University

**Chen Wang** [†]
Snap Inc.

**Aaron Sarna** [‡]
Google Research

**Yonglong Tian** [†]
MIT

**Phillip Isola** [†]
MIT

**Aaron Maschinot**
Google Research

**Ce Liu**
Google Research

**Dilip Krishnan**
Google Research

## Abstract

Contrastive learning applied to self-supervised representation learning has seen a resurgence in recent years, leading to state of the art performance in the unsupervised training of deep image models. Modern batch contrastive approaches subsume or significantly outperform traditional contrastive losses such as triplet, max-margin and the N-pairs loss. In this work, we extend the self-supervised batch contrastive approach to the *fully-supervised* setting, allowing us to effectively leverage label information. Clusters of points belonging to the same class are pulled together in embedding space, while simultaneously pushing apart clusters of samples from different classes. We analyze two possible versions of the supervised contrastive (SupCon) loss, identifying the best-performing formulation of the loss. On ResNet-200, we achieve top-1 accuracy of $81.4\%$ on the ImageNet dataset, which is $0.8\%$ above the best number reported for this architecture. We show consistent outperformance over cross-entropy on other datasets and two ResNet variants. The loss shows benefits for robustness to natural corruptions, and is more stable to hyperparameter settings such as optimizers and data augmentations. Our loss function is simple to implement and reference TensorFlow code is released at https://t.ly/supcon [1].

## 1 Introduction

The cross-entropy loss is the most widely used loss function for supervised learning of deep classification models. A number of works have explored shortcomings of this loss, such as lack of robustness to noisy labels [63, 46] and the possibility of poor margins [10, 31], leading to reduced generalization performance. However, in practice, most proposed alternatives have not worked better for large-scale datasets, such as ImageNet [7], as evidenced by the continued use of cross-entropy to achieve state of the art results [5, 6, 55, 25].

In recent years, a resurgence of work in contrastive learning has led to major advances in self-supervised

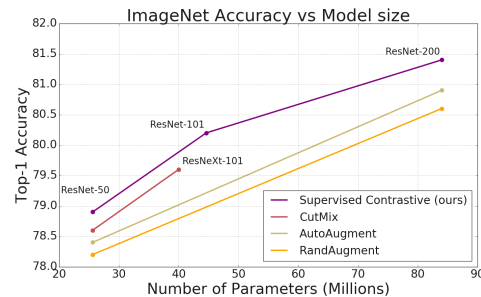

Figure 1: Our SupCon loss consistently outperforms cross-entropy with standard data augmentations. We show top-1 accuracy for the ImageNet dataset, on ResNet-50, ResNet-101 and ResNet-200, and compare against AutoAugment [5], RandAugment [6] and CutMix [59].

---

[*] Equal contribution.

[†] Work done while at Google Research.

[‡] Corresponding author: sarna@google.com

[1] PyTorch implementation: https://github.com/HobbitLong/SupContrast

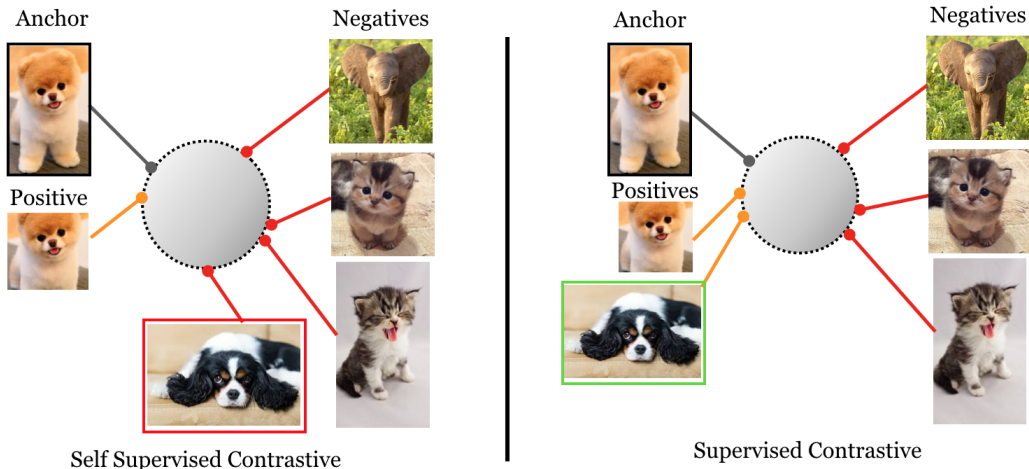

Figure 2: Supervised vs. self-supervised contrastive losses: The self-supervised contrastive loss (left, Eq. 1) contrasts a *single* positive for each anchor (i.e., an augmented version of the same image) against a set of negatives consisting of the entire remainder of the batch. The supervised contrastive loss (right) considered in this paper (Eq. 2), however, contrasts the set of *all* samples from the same class as positives against the negatives from the remainder of the batch. As demonstrated by the photo of the black and white puppy, taking class label information into account results in an embedding space where elements of the same class are more closely aligned than in the self-supervised case.

representation learning [54, 18, 38, 48, 22, 3, 15]. The common idea in these works is the following: pull together an anchor and a "positive" sample in embedding space, and push apart the anchor from many "negative" samples. Since no labels are available, a positive pair often consists of data augmentations of the sample, and negative pairs are formed by the anchor and randomly chosen samples from the minibatch. This is depicted in Fig. 2 (left). In [38, 48], connections are made of the contrastive loss to maximization of mutual information between different views of the data.

In this work, we propose a loss for supervised learning that builds on the contrastive self-supervised literature by leveraging label information. Normalized embeddings from the *same class* are pulled closer together than embeddings from *different classes*. Our technical novelty in this work is to consider *many positives* per anchor in addition to many negatives (as opposed to self-supervised contrastive learning which uses only a single positive). These positives are drawn from samples of the same class as the anchor, rather than being data augmentations of the anchor, as done in self-supervised learning. While this is a simple extension to the self-supervised setup, it is non-obvious how to setup the loss function correctly, and we analyze two alternatives. Fig. 2 (right) and Fig. 1 (Supplementary) provide a visual explanation of our proposed loss. Our loss can be seen as a generalization of both the triplet [52] and N-pair losses [45]; the former uses only one positive and one negative sample per anchor, and the latter uses one positive and many negatives. The use of many positives and many negatives for each anchor allows us to achieve state of the art performance without the need for hard negative mining, which can be difficult to tune properly. To the best of our knowledge, this is the first contrastive loss to consistently perform better than cross-entropy on large-scale classification problems. Furthermore, it provides a unifying loss function that can be used for either self-supervised or supervised learning.

Our resulting loss, SupCon, is simple to implement and stable to train, as our empirical results show. It achieves excellent top-1 accuracy on the ImageNet dataset on the ResNet-50 and ResNet-200 architectures [17]. On ResNet-200 [5], we achieve a top-1 accuracy of $81.4\%$, which is a $0.8\%$ improvement over the state of the art [30] cross-entropy loss on the same architecture (see Fig. 1). The gain in top-1 accuracy is accompanied by increased robustness as measured on the ImageNet-C dataset [19]. Our main contributions are summarized below:

1. We propose a novel extension to the contrastive loss function that allows for multiple positives per anchor, thus adapting contrastive learning to the fully supervised setting. Analytically and empirically, we show that a naïve extension performs much worse than our proposed version.

2. We show that our loss provides consistent boosts in top-1 accuracy for a number of datasets. It is also more robust to natural corruptions.
3. We demonstrate analytically that the gradient of our loss function encourages learning from hard positives and hard negatives.
4. We show empirically that our loss is less sensitive than cross-entropy to a range of hyperparameters.

## 2   Related Work

Our work draws on existing literature in self-supervised representation learning, metric learning and supervised learning. Here we focus on the most relevant papers. The cross-entropy loss was introduced as a powerful loss function to train deep networks [40, 1, 29]. The key idea is simple and intuitive: each class is assigned a target (usually 1-hot) vector. However, it is unclear why these target labels should be the optimal ones and some work has tried to identify better target label vectors, e.g. [56]. A number of papers have studied other drawbacks of the cross-entropy loss, such as sensitivity to noisy labels [63, 46], presence of adversarial examples [10, 36], and poor margins [2]. Alternative losses have been proposed, but the most effective ideas in practice have been approaches that change the reference label distribution, such as label smoothing [47, 35], data augmentations such as Mixup [60] and CutMix [59], and knowledge distillation [21].

Powerful self-supervised representation learning approaches based on deep learning models have recently been developed in the natural language domain [8, 57, 33]. In the image domain, pixel-predictive approaches have also been used to learn embeddings [9, 61, 62, 37]. These methods try to predict missing parts of the input signal. However, a more effective approach has been to replace a dense per-pixel predictive loss, with a loss in lower-dimensional representation space. The state of the art family of models for self-supervised representation learning using this paradigm are collected under the umbrella of contrastive learning [54, 18, 22, 48, 43, 3, 50]. In these works, the losses are inspired by noise contrastive estimation [13, 34] or N-pair losses [45]. Typically, the loss is applied at the last layer of a deep network. At test time, the embeddings from a previous layer are utilized for downstream transfer tasks, fine tuning or direct retrieval tasks. [15] introduces the approximation of only back-propagating through part of the loss, and also the approximation of using stale representations in the form of a memory bank.

Closely related to contrastive learning is the family of losses based on metric distance learning or triplets [4, 52, 42]. These losses have been used to learn powerful representations, often in supervised settings, where labels are used to guide the choice of positive and negative pairs. The key distinction between triplet losses and contrastive losses is the number of positive and negative pairs per data point; triplet losses use exactly one positive and one negative pair per anchor. In the supervised metric learning setting, the positive pair is chosen from the same class and the negative pair is chosen from other classes, nearly always requiring hard-negative mining for good performance [42]. Self-supervised contrastive losses similarly use just one positive pair for each anchor sample, selected using either co-occurrence [18, 22, 48] or data augmentation [3]. The major difference is that many negative pairs are used for each anchor. These are usually chosen uniformly at random using some form of weak knowledge, such as patches from other images, or frames from other randomly chosen videos, relying on the assumption that this approach yields a very low probability of false negatives.

Resembling our supervised contrastive approach is the soft-nearest neighbors loss introduced in [41] and used in [53]. Like [53], we improve upon [41] by normalizing the embeddings and replacing euclidean distance with inner products. We further improve on [53] by the increased use of data augmentation, a disposable contrastive head and two-stage training (contrastive followed by cross-entropy), and crucially, changing the form of the loss function to significantly improve results (see Section 3). [12] also uses a closely related loss formulation to ours to *entangle* representations at intermediate layers by maximizing the loss. Most similar to our method is the Compact Clustering via Label Propagation (CCLP) regularizer in Kamnitsas et. al. [24]. While CCLP focuses mostly on the semi-supervised case, in the fully supervised case the regularizer reduces to almost exactly our loss formulation. Important practical differences include our normalization of the contrastive embedding onto the unit sphere, tuning of a temperature parameter in the contrastive objective, and stronger augmentation. Additionally, Kamnitsas et. al. use the contrastive embedding as an input to a classification head, which is trained jointly with the CCLP regularizer, while SupCon employs a

two stage training and discards the contrastive head. Lastly, the scale of experiments in Kamnitsas et. al. is much smaller than in this work. Merging the findings of our paper and CCLP is a promising direction for semi-supervised learning research.

# 3 Method

Our method is structurally similar to that used in [48, 3] for self-supervised contrastive learning, with modifications for supervised classification. Given an input batch of data, we first apply data augmentation twice to obtain two copies of the batch. Both copies are forward propagated through the encoder network to obtain a 2048-dimensional normalized embedding. During training, this representation is further propagated through a projection network that is discarded at inference time. The supervised contrastive loss is computed on the outputs of the projection network. To use the trained model for classification, we train a linear classifier on top of the frozen representations using a cross-entropy loss. Fig. 1 in the Supplementary material provides a visual explanation.

## 3.1 Representation Learning Framework

The main components of our framework are:

- *Data Augmentation* module, $Aug(\cdot)$. For each input sample, $\boldsymbol{x}$, we generate two random augmentations, $\tilde{\boldsymbol{x}} = Aug(\boldsymbol{x})$, each of which represents a different *view* of the data and contains some subset of the information in the original sample. Sec. 4 gives details of the augmentations.

- *Encoder Network*, $Enc(\cdot)$, which maps $\boldsymbol{x}$ to a representation vector, $\boldsymbol{r} = Enc(\boldsymbol{x}) \in \mathcal{R}^{D_E}$. Both augmented samples are separately input to the same encoder, resulting in a pair of representation vectors. $\boldsymbol{r}$ is normalized to the unit hypersphere in $\mathcal{R}^{D_E}$ ($D_E = 2048$ in all our experiments in the paper). Consistent with the findings of [42, 51], our analysis and experiments show that this normalization improves top-1 accuracy.

- *Projection Network*, $Proj(\cdot)$, which maps $\boldsymbol{r}$ to a vector $\boldsymbol{z} = Proj(\boldsymbol{r}) \in \mathcal{R}^{D_P}$. We instantiate $Proj(\cdot)$ as either a multi-layer perceptron [14] with a single hidden layer of size 2048 and output vector of size $D_P = 128$ or just a single linear layer of size $D_P = 128$; we leave to future work the investigation of optimal $Proj(\cdot)$ architectures. We again normalize the output of this network to lie on the unit hypersphere, which enables using an inner product to measure distances in the projection space. As in self-supervised contrastive learning [48, 3], we discard $Proj(\cdot)$ at the end of contrastive training. As a result, our inference-time models contain exactly the same number of parameters as a cross-entropy model using the same encoder, $Enc(\cdot)$.

## 3.2 Contrastive Loss Functions

Given this framework, we now look at the family of contrastive losses, starting from the self-supervised domain and analyzing the options for adapting it to the supervised domain, showing that one formulation is superior. For a set of $N$ randomly sampled sample/label pairs, $\{\boldsymbol{x}_k, \boldsymbol{y}_k\}_{k=1...N}$, the corresponding batch used for training consists of $2N$ pairs, $\{\tilde{\boldsymbol{x}}_\ell, \tilde{\boldsymbol{y}}_\ell\}_{\ell=1...2N}$, where $\tilde{\boldsymbol{x}}_{2k}$ and $\tilde{\boldsymbol{x}}_{2k-1}$ are two random augmentations (a.k.a., "views") of $\boldsymbol{x}_k$ ($k = 1...N$) and $\tilde{\boldsymbol{y}}_{2k-1} = \tilde{\boldsymbol{y}}_{2k} = \boldsymbol{y}_k$. For the remainder of this paper, we will refer to a set of $N$ samples as a "batch" and the set of $2N$ augmented samples as a "multiviewed batch".

### 3.2.1 Self-Supervised Contrastive Loss

Within a multiviewed batch, let $i \in I \equiv \{1...2N\}$ be the index of an arbitrary augmented sample, and let $j(i)$ be the index of the other augmented sample originating from the same source sample. In *self-supervised* contrastive learning (e.g., [3, 48, 18, 22]), the loss takes the following form.

$$\mathcal{L}^{self} = \sum_{i \in I} \mathcal{L}_i^{self} = -\sum_{i \in I} \log \frac{\exp\left(\boldsymbol{z}_i \cdot \boldsymbol{z}_{j(i)}/\tau\right)}{\sum\limits_{a \in A(i)} \exp\left(\boldsymbol{z}_i \cdot \boldsymbol{z}_a/\tau\right)} \tag{1}$$

Here, $\boldsymbol{z}_\ell = Proj(Enc(\tilde{\boldsymbol{x}}_\ell)) \in \mathcal{R}^{D_P}$, the $\cdot$ symbol denotes the inner (dot) product, $\tau \in \mathcal{R}^+$ is a scalar temperature parameter, and $A(i) \equiv I \setminus \{i\}$. The index $i$ is called the *anchor*, index $j(i)$ is called the *positive*, and the other $2(N - 1)$ indices ($\{k \in A(i) \setminus \{j(i)\}\}$) are called the *negatives*.

Note that for each anchor $i$, there is 1 positive pair and $2N - 2$ negative pairs. The denominator has a total of $2N - 1$ terms (the positive and negatives).

### 3.2.2 Supervised Contrastive Losses

For supervised learning, the contrastive loss in Eq. 1 is incapable of handling the case where, due to the presence of labels, more than one sample is known to belong to the same class. Generalization to an arbitrary numbers of positives, though, leads to a choice between multiple possible functions. Eqs. 2 and 3 present the two most straightforward ways to generalize Eq. 1 to incorporate supervision.

$$\mathcal{L}_{out}^{sup} = \sum_{i \in I} \mathcal{L}_{out,i}^{sup} = \sum_{i \in I} \frac{-1}{|P(i)|} \sum_{p \in P(i)} \log \frac{\exp\left(\boldsymbol{z}_i \boldsymbol{\cdot} \boldsymbol{z}_p / \tau\right)}{\sum\limits_{a \in A(i)} \exp\left(\boldsymbol{z}_i \boldsymbol{\cdot} \boldsymbol{z}_a / \tau\right)} \tag{2}$$

$$\mathcal{L}_{in}^{sup} = \sum_{i \in I} \mathcal{L}_{in,i}^{sup} = \sum_{i \in I} -\log \left\{ \frac{1}{|P(i)|} \sum_{p \in P(i)} \frac{\exp\left(\boldsymbol{z}_i \boldsymbol{\cdot} \boldsymbol{z}_p / \tau\right)}{\sum\limits_{a \in A(i)} \exp\left(\boldsymbol{z}_i \boldsymbol{\cdot} \boldsymbol{z}_a / \tau\right)} \right\} \tag{3}$$

Here, $P(i) \equiv \{p \in A(i) : \tilde{\boldsymbol{y}}_p = \tilde{\boldsymbol{y}}_i\}$ is the set of indices of all positives in the multiviewed batch distinct from $i$, and $|P(i)|$ is its cardinality. In Eq. 2, the summation over positives is located *outside* of the log ($\mathcal{L}_{out}^{sup}$) while in Eq. 3, the summation is located *inside* of the log ($\mathcal{L}_{in}^{sup}$). Both losses have the following desirable properties:

- **Generalization to an arbitrary number of positives.** The major structural change of Eqs. 2 and 3 over Eq. 1 is that now, for any anchor, *all* positives in a multiviewed batch (i.e., the augmentation-based sample as well as any of the remaining samples with the same label) contribute to the numerator. For randomly-generated batches whose size is large with respect to the number of classes, multiple additional terms will be present (on average, $N/C$, where $C$ is the number of classes). The supervised losses encourage the encoder to give closely aligned representations to *all* entries from the same class, resulting in a more robust clustering of the representation space than that generated from Eq. 1, as is supported by our experiments in Sec. 4.

- **Contrastive power increases with more negatives.** Eqs. 2 and 3 both preserve the summation over negatives in the contrastive denominator of Eq. 1. This form is largely motivated by noise contrastive estimation and N-pair losses [13, 45], wherein the ability to discriminate between signal and noise (negatives) is improved by adding more examples of negatives. This property is important for representation learning via self-supervised contrastive learning, with many papers showing increased performance with increasing number of negatives [18, 15, 48, 3].

- **Intrinsic ability to perform hard positive/negative mining.** When used with *normalized* representations, the loss in Eq. 1 induces a gradient structure that gives rise to implicit hard positive/negative mining. The gradient contributions from *hard* positives/negatives (i.e., ones against which continuing to contrast the anchor *greatly* benefits the encoder) are large while those for *easy* positives/negatives (i.e., ones against which continuing to contrast the anchor only *weakly* benefits the encoder) are small. Furthermore, for hard positives, the effect increases (asymptotically) as the number of negatives does. Eqs. 2 and 3 both preserve this useful property and generalize it to all positives. This implicit property allows the contrastive loss to sidestep the need for explicit hard mining, which is a delicate but critical part of many losses, such as triplet loss [42]. We note that this implicit property applies to both supervised and self-supervised contrastive losses, but our derivation is the first to clearly show this property. We provide a full derivation of this property from the loss gradient in the Supplementary material.

The two loss formulations are not, however, equivalent. Because $\log$ is a concave function, Jensen's Inequality [23] implies that $\mathcal{L}_{out}^{sup} \leq \mathcal{L}_{in}^{sup}$. One might thus be tempted to conclude that $\mathcal{L}_{in}^{sup}$ is the superior supervised loss function (since it bounds $\mathcal{L}_{out}^{sup}$). However, this conclusion is *not* supported analytically. Table 1 compares the ImageNet [7] top-1 classification accuracy using $\mathcal{L}_{out}^{sup}$ and $\mathcal{L}_{in}^{sup}$ for different batch sizes ($N$) on the ResNet-50 [17] architecture. The $\mathcal{L}_{out}^{sup}$ supervised

| Loss | Top-1 |
|---|---|
| $\mathcal{L}_{out}^{sup}$ | 78.7% |
| $\mathcal{L}_{in}^{sup}$ | 67.4% |

Table 1: ImageNet Top-1 classification accuracy for supervised contrastive losses on ResNet-50 for a batch size of 6144.

loss achieves significantly higher performance than $\mathcal{L}_{in}^{sup}$. We conjecture that this is due to the gradient of $\mathcal{L}_{in}^{sup}$ having structure less optimal for training than that of $\mathcal{L}_{out}^{sup}$. For $\mathcal{L}_{out}^{sup}$, the positives

normalization factor (i.e., $1/|P(i)|$) serves to remove bias present in the positives in a multiviewed batch contributing to the loss. However, though $\mathcal{L}_{in}^{sup}$ also contains the same normalization factor, it is located *inside* of the log. It thus contributes only an additive constant to the overall loss, which does not affect the gradient. Without any normalization effects, the gradients of $\mathcal{L}_{in}^{sup}$ are more susceptible to bias in the positives, leading to sub-optimal training.

An analysis of the gradients themselves supports this conclusion. As shown in the Supplementary, the gradient for *either* $\mathcal{L}_{out,i}^{sup}$ or $\mathcal{L}_{in,i}^{sup}$ with respect to the embedding $z_i$ has the following form.

$$\frac{\partial \mathcal{L}_i^{sup}}{\partial z_i} = \frac{1}{\tau} \left\{ \sum_{p \in P(i)} z_p (P_{ip} - X_{ip}) + \sum_{n \in N(i)} z_n P_{in} \right\} \tag{4}$$

Here, $N(i) \equiv \{n \in A(i) : \tilde{y}_n \neq \tilde{y}_i\}$ is the set of indices of all negatives in the multiviewed batch, and $P_{ix} \equiv \exp\left(z_i \cdot z_x / \tau\right) / \sum_{a \in A(i)} \exp\left(z_i \cdot z_a / \tau\right)$. The difference between the gradients for the two losses is in $X_{ip}$.

$$X_{ip} = \begin{cases} \frac{\exp(z_i \cdot z_p / \tau)}{\sum\limits_{p' \in P(i)} \exp\left(z_i \cdot z_{p'} / \tau\right)} & , \quad \text{if } \mathcal{L}_i^{sup} = \mathcal{L}_{in,i}^{sup} \\ \frac{1}{|P(i)|} & , \quad \text{if } \mathcal{L}_i^{sup} = \mathcal{L}_{out,i}^{sup} \end{cases} \tag{5}$$

If each $z_p$ is set to the (less biased) mean positive representation vector, $\overline{z}$, $X_{ip}^{in}$ reduces to $X_{ip}^{out}$:

$$X_{ip}^{in}\Big|_{z_p = \overline{z}} = \frac{\exp\left(z_i \cdot \overline{z}/\tau\right)}{\sum\limits_{p' \in P(i)} \exp\left(z_i \cdot \overline{z}/\tau\right)} = \frac{\exp\left(z_i \cdot \overline{z}/\tau\right)}{|P(i)| \cdot \exp\left(z_i \cdot \overline{z}/\tau\right)} = \frac{1}{|P(i)|} = X_{ip}^{out} \tag{6}$$

From the form of $\partial \mathcal{L}_i^{sup} / \partial z_i$, we conclude that the stabilization due to using the mean of positives benefits training. Throughout the rest of the paper, we consider only $\mathcal{L}_{out}^{sup}$.

### 3.2.3 Connection to Triplet Loss and N-pairs Loss

Supervised contrastive learning is closely related to the triplet loss [52], one of the widely-used loss functions for supervised learning. In the Supplementary, we show that the triplet loss is a special case of the contrastive loss when one positive and one negative are used. When more than one negative is used, we show that the SupCon loss becomes equivalent to the N-pairs loss [45].

## 4 Experiments

We evaluate our SupCon loss ($\mathcal{L}_{out}^{sup}$, Eq. 2) by measuring classification accuracy on a number of common image classification benchmarks including CIFAR-10 and CIFAR-100 [27] and ImageNet [7]. We also benchmark our ImageNet models on robustness to common image corruptions [19] and show how performance varies with changes to hyperparameters and reduced data. For the encoder network ($Enc(\cdot)$) we experimented with three commonly used encoder architectures: ResNet-50, ResNet-101, and ResNet-200 [17]. The normalized activations of the final pooling layer ($D_E = 2048$) are used as the representation vector. We experimented with four different implementations of the $Aug(\cdot)$ data augmentation module: AutoAugment [5]; RandAugment [6]; SimAugment [3], and Stacked RandAugment [49] (see details of our SimAugment and Stacked RandAugment implementations in the Supplementary). AutoAugment outperforms all other data augmentation strategies on ResNet-50 for both SupCon and cross-entropy. Stacked RandAugment performed best for ResNet-200 for both loss functions. We provide more details in the Supplementary.

### 4.1 Classification Accuracy

Table 2 shows that SupCon generalizes better than cross-entropy, margin classifiers (with use of labels) and unsupervised contrastive learning techniques on CIFAR-10, CIFAR-100 and ImageNet datasets. Table 3 shows results for ResNet-50 and ResNet-200 (we use ResNet-v1 [17]) for ImageNet. We achieve a new state of the art accuracy of 78.7% on ResNet-50 with AutoAugment (for comparison, a number of the other top-performing methods are shown in Fig. 1). Note that we also

| Dataset | SimCLR[3] | Cross-Entropy | Max-Margin [32] | SupCon |
|---------|-----------|---------------|-----------------|--------|
| CIFAR10 | 93.6 | 95.0 | 92.4 | **96.0** |
| CIFAR100 | 70.7 | 75.3 | 70.5 | **76.5** |
| ImageNet | 70.2 | 78.2 | 78.0 | **78.7** |

Table 2: Top-1 classification accuracy on ResNet-50 [17] for various datasets. We compare cross-entropy training, unsupervised representation learning (SimCLR [3]), max-margin classifiers [32] and SupCon (ours). We re-implemented and tuned hyperparameters for all baseline numbers except margin classifiers where we report published results. Note that the CIFAR-10 and CIFAR-100 results are from our PyTorch implementation and ImageNet from our TensorFlow implementation.

| Loss | Architecture | Augmentation | Top-1 | Top-5 |
|------|--------------|--------------|-------|-------|
| Cross-Entropy (baseline) | ResNet-50 | MixUp [60] | 77.4 | 93.6 |
| Cross-Entropy (baseline) | ResNet-50 | CutMix [59] | 78.6 | 94.1 |
| Cross-Entropy (baseline) | ResNet-50 | AutoAugment [5] | 78.2 | 92.9 |
| Cross-Entropy (our impl.) | ResNet-50 | AutoAugment [30] | 77.6 | 95.3 |
| SupCon | ResNet-50 | AutoAugment [5] | **78.7** | **94.3** |
| Cross-Entropy (baseline) | ResNet-200 | AutoAugment [5] | 80.6 | 95.3 |
| Cross-Entropy (our impl.) | ResNet-200 | Stacked RandAugment [49] | 80.9 | 95.2 |
| SupCon | ResNet-200 | Stacked RandAugment [49] | **81.4** | **95.9** |
| SupCon | ResNet-101 | Stacked RandAugment [49] | 80.2 | 94.7 |

Table 3: Top-1/Top-5 accuracy results on ImageNet for AutoAugment [5] with ResNet-50 and for Stacked RandAugment [49] with ResNet-101 and ResNet-200. The baseline numbers are taken from the referenced papers, and we also re-implement cross-entropy.

achieve a slight improvement over CutMix [59], which is considered to be a state of the art data augmentation strategy. Incorporating data augmentation strategies such as CutMix [59] and MixUp [60] into contrastive learning could potentially improve results further.

We also experimented with memory based alternatives [15]. On ImageNet, with a memory size of 8192 (requiring only the storage of 128-dimensional vectors), a batch size of 256, and SGD optimizer, running on 8 Nvidia V100 GPUs, SupCon is able to achieve 79.1% top-1 accuracy on ResNet-50. This is in fact slightly better than the 78.7% accuracy with 6144 batch size (and no memory); and with significantly reduced compute and memory footprint.

Since SupCon uses 2 views per sample, its batch sizes are effectively twice the cross-entropy equivalent. We therefore also experimented with the cross-entropy ResNet-50 baselines using a batch size of 12,288. These only achieved 77.5% top-1 accuracy. We additionally experimented with increasing the number of training epochs for cross-entropy all the way to 1400, but this actually decreased accuracy (77.0%).

We tested the N-pairs loss [45] in our framework with a batch size of 6144. N-pairs achieves only 57.4% top-1 accuracy on ImageNet. We believe this is due to multiple factors missing from N-pairs loss compared to supervised contrastive: the use of multiple views; lower temperature; and many more positives. We show some results of the impact of the number of positives per anchor in the Supplementary (Sec. 6), and the N-pairs result is inline with them. We also note that the original N-pairs paper [45] has already shown the outperformance of N-pairs loss to triplet loss.

### 4.2 Robustness to Image Corruptions and Reduced Training Data

Deep neural networks lack robustness to out of distribution data or natural corruptions such as noise, blur and JPEG compression. The benchmark ImageNet-C dataset [19] is used to measure trained model performance on such corruptions. In Fig. 3(left), we compare the supervised contrastive models to cross-entropy using the Mean Corruption Error (mCE) and Relative Mean Corruption Error metrics [19]. Both metrics measure average degradation in performance compared to ImageNet test set, averaged over all possible corruptions and severity levels. Relative mCE is a better metric when we compare models with different Top-1 accuracy, while mCE is a better measure of absolute robustness to corruptions. The SupCon models have lower mCE values across different corruptions, showing increased robustness. We also see from Fig. 3(right) that SupCon models demonstrate lesser degradation in accuracy with increasing corruption severity.

| Loss | Architecture | rel. mCE | mCE |
|---|---|---|---|
| Cross-Entropy (baselines) | AlexNet [28] | 100.0 | 100.0 |
| | VGG-19+BN [44] | 122.9 | 81.6 |
| | ResNet-18 [17] | 103.9 | 84.7 |
| Cross-Entropy (our implementation) | ResNet-50 | 96.2 | 68.6 |
| | ResNet-200 | 69.1 | 52.4 |
| Supervised Contrastive | ResNet-50 | **94.6** | **67.2** |
| | ResNet-200 | **66.5** | **50.6** |

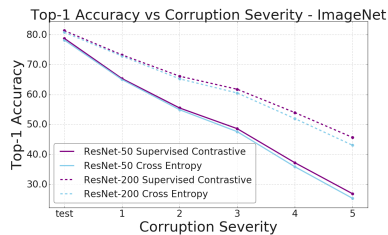

Figure 3: Training with supervised contrastive loss makes models more robust to corruptions in images. **Left**: Robustness as measured by Mean Corruption Error (mCE) and relative mCE over the ImageNet-C dataset [19] (lower is better). **Right**: Mean Accuracy as a function of corruption severity averaged over all various corruptions. (higher is better).

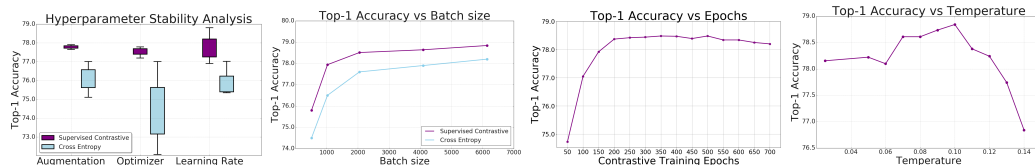

Figure 4: Accuracy of cross-entropy and supervised contrastive loss as a function of hyperparameters and training data size, all measured on ImageNet with a ResNet-50 encoder. (From left to right) **(a)**: Standard boxplot showing Top-1 accuracy vs changes in augmentation, optimizer and learning rates. **(b)**: Top-1 accuracy as a function of batch size shows both losses benefit from larger batch sizes while Supervised Contrastive has higher Top-1 accuracy even when trained with smaller batch sizes. **(c)**: Top-1 accuracy as a function of SupCon pretraining epochs. **(d)**: Top-1 accuracy as a function of temperature during pretraining stage for SupCon.

| | Food | CIFAR10 | CIFAR100 | Birdsnap | SUN397 | Cars | Aircraft | VOC2007 | DTD | Pets | Caltech-101 | Flowers | Mean |
|---|---|---|---|---|---|---|---|---|---|---|---|---|---|
| SimCLR-50 [3] | **88.20** | **97.70** | **85.90** | **75.90** | **63.50** | 91.30 | **88.10** | 84.10 | 73.20 | 89.20 | 92.10 | **97.00** | **84.81** |
| Xent-50 | 87.38 | 96.50 | 84.93 | 74.70 | 63.15 | 89.57 | 80.80 | **85.36** | **76.86** | 92.35 | **92.34** | 96.93 | **84.67** |
| SupCon-50 | 87.23 | 97.42 | 84.27 | 75.15 | 58.04 | **91.69** | 84.09 | 85.17 | 74.60 | **93.47** | 91.04 | 96.0 | **84.27** |
| Xent-200 | **89.36** | 97.96 | 86.49 | **76.50** | **64.36** | 90.01 | 84.22 | **86.27** | **76.76** | **93.48** | 93.84 | **97.20** | **85.77** |
| SupCon-200 | 88.62 | **98.28** | **87.28** | 76.26 | 60.46 | **91.78** | **88.68** | 85.18 | 74.26 | 93.12 | **94.91** | 96.97 | **85.67** |

Table 4: Transfer learning results. Numbers are mAP for VOC2007 [11]; mean-per-class accuracy for Aircraft, Pets, Caltech, and Flowers; and top-1 accuracy for all other datasets.

## 4.3 Hyperparameter Stability

We experimented with hyperparameter stability by changing augmentations, optimizers and learning rates one at a time from the best combination for each of the methodologies. In Fig. 4(a), we compare the top-1 accuracy of SupCon loss against cross-entropy across changes in augmentations (RandAugment [6], AutoAugment [5], SimAugment [3], Stacked RandAugment [49]); optimizers (LARS, SGD with Momentum and RMSProp); and learning rates. We observe significantly lower variance in the output of the contrastive loss. Note that batch sizes for cross-entropy and supervised contrastive are the same, thus ruling out any batch-size effects. In Fig. 4(b), sweeping batch size and holding all other hyperparameters constant results in consistently better top-1 accuracy of the supervised contrastive loss.

## 4.4 Transfer Learning

We evaluate the learned representation for fine-tuning on 12 natural image datasets, following the protocol in Chen et.al. [3]. SupCon is on par with cross-entropy and *self-supervised* contrastive loss on transfer learning performance when trained on the same architecture (Table 4). Our results are consistent with the findings in [16] and [26]: while better ImageNet models are correlated with better transfer performance, the dominant factor is architecture. Understanding the connection between training objective and transfer performance is left to future work.

## 4.5   Training Details

The SupCon loss was trained for 700 epochs during pretraining for ResNet-200 and 350 epochs for smaller models. Fig. 4(c) shows accuracy as a function of SupCon training epochs for a ResNet50, demonstrating that even 200 epochs is likely sufficient for most purposes.

An (optional) additional step of training a linear classifier is used to compute top-1 accuracy. This is not needed if the purpose is to use representations for transfer learning tasks or retrieval. The second stage needs as few as 10 epochs of additional training. Note that in practice the linear classifier can be trained jointly with the encoder and projection networks by blocking gradient propagation from the linear classifier back to the encoder, and achieve roughly the same results without requiring two-stage training. We chose not to do that here to help isolate the effects of the SupCon loss.

We trained our models with batch sizes of up to 6144, although batch sizes of 2048 suffice for most purposes for both SupCon and cross-entropy losses (as shown in Fig. 4(b)). We associate some of the performance increase with batch size to the effect on the gradient due to hard positives increasing with an increasing number of negatives (see the Supplementary for details). We report metrics for experiments with batch size 6144 for ResNet-50 and batch size 4096 for ResNet-200 (due to the larger network size, a smaller batch size is necessary). We observed that for a fixed batch size it was possible to train with SupCon using larger learning rates than what was required by cross-entropy to achieve similar performance.

All our results used a temperature of $\tau = 0.1$. Smaller temperature benefits training more than higher ones, but extremely low temperatures are harder to train due to numerical instability. Fig. 4(d) shows the effect of temperature on Top-1 performance of supervised contrastive learning. As we can see from Eq. 4, the gradient scales inversely with choice of temperature $\tau$; therefore we rescale the loss by $\tau$ during training for stability.

We experimented with standard optimizers such as LARS [58], RMSProp [20] and SGD with momentum [39] in different permutations for the initial pre-training step and training of the dense layer. While SGD with momentum works best for training ResNets with cross-entropy, we get the best performance for SupCon on ImageNet by using LARS for pre-training and RMSProp to training the linear layer. For CIFAR10 and CIFAR100 SGD with momentum performed best. Additional results for combinations of optimizers are provided in the Supplementary. Reference code is released at https://t.ly/supcon.

# Broader Impact

This work provides a technical advancement in the field of supervised classification, which already has tremendous impact throughout industry. Whether or not they realize it, most people experience the results of this type of classifier many times a day.

As we have shown, supervised contrastive learning can improve both the accuracy and robustness of classifiers, which for most applications should strictly be an improvement. For example, an autonomous car that makes a classification error due to data distribution shift can result in catastrophic results. Thus decreasing this class of error undoubtedly promotes safety. Human driver error is a massive source of fatalities around the world, so improving the safety of autonomous cars furthers the efforts of replacing human drivers. The flip side of that progress is of course the potential for loss of employment in fields like trucking and taxi driving. Similar two-sided coins can be considered for assessing the impact of any application of classification.

An additional potential impact of our work in particular is showing the value of training with large batch sizes. Generally, large batch size training comes at the cost of substantial energy consumption, which unfortunately today requires the burning of fossil fuels, which in turn warms our planet. We are proud to say that the model training that was done in the course of this research was entirely carbon-neutral, where all power consumed was either green to start with, or offset by purchases of green energy. There is unfortunately no way to guarantee that once this research is publicly available that all practitioners of it will choose, or even have the ability to choose, to limit the environmental impact of their model training.

## Acknowledgments and Disclosure of Funding

Additional revenues related to this work: In the past 36 months, Phillip Isola has had employment at MIT, Google, and OpenAI; honorarium for lecturing at the ACDL summer school in Italy; honorarium for speaking at GIST AI Day in South Korea. P.I.'s lab at MIT has been supported by grants from Facebook, IBM, and the US Air Force; start up funding from iFlyTech via MIT; gifts from Adobe and Google; compute credit donations from Google Cloud.

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
