[Supplementary Material]

# Supervised Contrastive Learning - Supplementary Material

**Prannay Khosla** *     **Piotr Teterwak** *     **Chen Wang** †     **Aaron Sarna** ‡     **Yonglong Tian**
Citadel               Boston University        Snap Inc.           Google Research          MIT

**Phillip Isola**        **Aaron Maschinot**           **Ce Liu**            **Dilip Krishnan**
MIT                   Google Research            Google Research         Google Research

## 1   Training Setup

In Fig. 1, we compare the training setup for the cross-entropy, self-supervised contrastive and supervised contrastive (SupCon) losses. Note that the number of parameters in the inference models always stays the same. We also note that it is not necessary to train a linear classifier in the second stage, and previous works have used k-Nearest Neighbor classification [12] or prototype classification to evaluate representations on classification tasks. The linear classifier can also be trained jointly with the encoder, as long as it doesn't propagate gradients back to the encoder.

(a) Supervised Cross Entropy          (b) Self Supervised Contrastive          (c) Supervised Contrastive

Figure 1: Cross entropy, self-supervised contrastive loss and supervised contrastive loss: The cross entropy loss (left) uses labels and a softmax loss to train a classifier; the self-supervised contrastive loss (middle) uses a contrastive loss and data augmentations to learn representations. The supervised contrastive loss (right) also learns representations using a contrastive loss, but uses label information to sample positives in addition to augmentations of the same image. Both contrastive methods can have an optional second stage which trains a model on top of the learned representations.

## 2    Gradient Derivation

In Sec. 3 of the paper, we make the claim that the gradients of the two considered supervised contrastive losses, $\mathcal{L}_{out}^{sup}$ and $\mathcal{L}_{in}^{sup}$, with respect to a normalized projection network representation, $z_i$, have a nearly identical mathematical form. In this section, we perform derivations to show this is true. It is sufficient to show that this claim is true for $\mathcal{L}_{out,i}^{sup}$ and $\mathcal{L}_{in,i}^{sup}$. For convenience, we reprint below the expressions for each.

$$\mathcal{L}_{in,i}^{sup} = -\log \left\{ \frac{1}{|P(i)|} \sum_{p \in P(i)} \frac{\exp\left(z_i \cdot z_p / \tau\right)}{\sum\limits_{a \in A(i)} \exp\left(z_i \cdot z_a / \tau\right)} \right\} \tag{1}$$

$$\mathcal{L}_{out,i}^{sup} = \frac{-1}{|P(i)|} \sum_{p \in P(i)} \log \frac{\exp\left(z_i \cdot z_p / \tau\right)}{\sum\limits_{a \in A(i)} \exp\left(z_i \cdot z_a / \tau\right)} \tag{2}$$

We start by deriving the gradient of $\mathcal{L}_{in}^{sup}$ (Eq. 1):

$$
\begin{aligned}
\frac{\partial \mathcal{L}_{in}^{sup}}{\partial z_i} &= -\frac{\partial}{\partial z_i} \log \left\{ \frac{1}{|P(i)|} \sum_{p \in P(i)} \frac{\exp\left(z_i \cdot z_p / \tau\right)}{\sum\limits_{a \in A(i)} \exp\left(z_i \cdot z_a / \tau\right)} \right\} \\
&= \frac{\partial}{\partial z_i} \log \sum_{a \in A(i)} \exp\left(z_i \cdot z_a / \tau\right) - \frac{\partial}{\partial z_i} \log \sum_{p \in P(i)} \exp\left(z_i \cdot z_p / \tau\right) \\
&= \frac{1}{\tau} \frac{\sum\limits_{a \in A(i)} z_a \exp\left(z_i \cdot z_a / \tau\right)}{\sum\limits_{a \in A(i)} \exp\left(z_i \cdot z_a / \tau\right)} - \frac{1}{\tau} \frac{\sum\limits_{p \in P(i)} z_p \exp\left(z_i \cdot z_p / \tau\right)}{\sum\limits_{p \in P(i)} \exp\left(z_i \cdot z_p / \tau\right)} \\
&= \frac{1}{\tau} \frac{\sum\limits_{p \in P(i)} z_p \exp\left(z_i \cdot z_p / \tau\right) + \sum\limits_{n \in N(i)} z_n \exp\left(z_i \cdot z_n / \tau\right)}{\sum\limits_{a \in A(i)} \exp\left(z_i \cdot z_a / \tau\right)} - \frac{1}{\tau} \frac{\sum\limits_{p \in P(i)} z_p \exp\left(z_i \cdot z_p / \tau\right)}{\sum\limits_{p \in P(i)} \exp\left(z_i \cdot z_p / \tau\right)} \\
&= \frac{1}{\tau} \left\{ \sum_{p \in P(i)} z_p (P_{ip} - X_{ip}^{in}) + \sum_{n \in N(i)} z_n P_{in} \right\}
\end{aligned}
\tag{3}
$$

where we have defined:

$$P_{ip} \equiv \frac{\exp\left(z_i \cdot z_p / \tau\right)}{\sum_{a \in A(i)} \exp\left(z_i \cdot z_a / \tau\right)} \tag{4}$$

$$X_{ip}^{in} \equiv \frac{\exp\left(z_i \cdot z_p / \tau\right)}{\sum\limits_{p' \in P(i)} \exp\left(z_i \cdot z_{p'} / \tau\right)} \tag{5}$$

Though similar in structure, $P_{ip}$ and $X_{ip}^{in}$ are fundamentally different: $P_{ip}$ is the likelihood for $z_p$ with respect to all positives and negatives, while $X_{ip}^{in}$ is that but with respect to only the positives. $P_{in}$ is analogous to $P_{ip}$ but defines the likelihood of $z_n$. In particular, $P_{ip} \leq X_{ip}^{in}$. We now derive

the gradient of Eq. 2:

$$\frac{\partial \mathcal{L}_{out}^{sup}}{\partial \boldsymbol{z}_i} = \frac{-1}{|P(i)|} \sum_{p \in P(i)} \frac{\partial}{\partial \boldsymbol{z}_i} \left\{ \frac{\boldsymbol{z}_i \cdot \boldsymbol{z}_p}{\tau} - \log \sum_{a \in A(i)} \exp\left(\boldsymbol{z}_i \cdot \boldsymbol{z}_a / \tau\right) \right\}$$

$$= \frac{-1}{\tau|P(i)|} \sum_{p \in P(i)} \left\{ \boldsymbol{z}_p - \frac{\sum\limits_{a \in A(i)} \boldsymbol{z}_a \exp\left(\boldsymbol{z}_i \cdot \boldsymbol{z}_a / \tau\right)}{\sum\limits_{a \in A(i)} \exp\left(\boldsymbol{z}_i \cdot \boldsymbol{z}_a / \tau\right)} \right\}$$

$$= \frac{-1}{\tau|P(i)|} \sum_{p \in P(i)} \left\{ \boldsymbol{z}_p - \sum_{p' \in P(i)} \boldsymbol{z}_{p'} P_{ip'} - \sum_{n \in N(i)} \boldsymbol{z}_n P_{in} \right\}$$

$$= \frac{-1}{\tau|P(i)|} \left\{ \sum_{p \in P(i)} \boldsymbol{z}_p - \sum_{p \in P(i)} \sum_{p' \in P(i)} \boldsymbol{z}_{p'} P_{ip'} - \sum_{p \in P(i)} \sum_{n \in N(i)} \boldsymbol{z}_n P_{in} \right\}$$

$$= \frac{-1}{\tau|P(i)|} \left\{ \sum_{p \in P(i)} \boldsymbol{z}_p - \sum_{p' \in P(i)} \sum_{p \in P(i)} \boldsymbol{z}_{p'} P_{ip'} - \sum_{n \in N(i)} \sum_{p \in P(i)} \boldsymbol{z}_n P_{in} \right\}$$

$$= \frac{-1}{\tau|P(i)|} \left\{ \sum_{p \in P(i)} \boldsymbol{z}_p - \sum_{p' \in P(i)} |P(i)| \boldsymbol{z}_{p'} P_{ip'} - \sum_{n \in N(i)} |P(i)| \boldsymbol{z}_n P_{in} \right\}$$

$$= \frac{-1}{\tau|P(i)|} \left\{ \sum_{p \in P(i)} \boldsymbol{z}_p - \sum_{p \in P(i)} |P(i)| \boldsymbol{z}_p P_{ip} - \sum_{n \in N(i)} |P(i)| \boldsymbol{z}_n P_{in} \right\}$$

$$= \frac{1}{\tau} \left\{ \sum_{p \in P(i)} \boldsymbol{z}_p (P_{ip} - X_{ip}^{out}) + \sum_{n \in N(i)} \boldsymbol{z}_n P_{in} \right\} \tag{6}$$

where we have defined:

$$X_{ip}^{out} \equiv \frac{1}{|P(i)|} \tag{7}$$

Thus, both gradients (Eqs. 3 and 6) have a very similar form and can be written collectively as:

$$\frac{\partial \mathcal{L}_i^{sup}}{\partial \boldsymbol{z}_i} = \frac{1}{\tau} \left\{ \sum_{p \in P(i)} \boldsymbol{z}_p (P_{ip} - X_{ip}) + \sum_{n \in N(i)} \boldsymbol{z}_n P_{in} \right\} \tag{8}$$

where:

$$X_{ip} \equiv \begin{cases} \frac{\exp(\boldsymbol{z}_i \cdot \boldsymbol{z}_p / \tau)}{\sum\limits_{p' \in P(i)} \exp(\boldsymbol{z}_i \cdot \boldsymbol{z}_{p'} / \tau)} & , \quad \text{if } \mathcal{L}_i^{sup} = \mathcal{L}_{in,i}^{sup} \\ \frac{1}{|P(i)|} & , \quad \text{if } \mathcal{L}_i^{sup} = \mathcal{L}_{out,i}^{sup} \end{cases} \tag{9}$$

This corresponds to Eq. 4 and subsequent analysis in the paper.

## 3 Intrinsic Hard Positive and Negative Mining Properties

The contrastive loss is structured so that gradients with respect to the *unnormalized* projection network representations provide an intrinsic mechanism for hard positive/negative mining during training. For losses such as the triplet loss or max-margin, hard mining is known to be crucial to their performance. For contrastive loss, we show analytically that hard mining is intrinsic and thus removes the need for complicated hard mining algorithms.

As shown in Sec. 2, the gradients of both $\mathcal{L}_{out}^{sup}$ and $\mathcal{L}_{in}^{sup}$ are given by Eq. 6. Additionally, note that the self-supervised contrastive loss, $\mathcal{L}_i^{self}$, is a special case of either of the two supervised contrastive losses (when $P(i) = j(i)$). So by showing that Eq. 6 has structure that provides hard

positive/negative mining, it will be shown to be true for all three contrastive losses (self-supervised and both supervised versions).

The projection network applies a normalization to its outputs[4]. We shall let $\boldsymbol{w}_i$ denote the projection network output *prior* to normalization, i.e., $\boldsymbol{z}_i = \boldsymbol{w}_i/\|\boldsymbol{w}_i\|$. As we show below, normalizing the representations provides structure (when combined with Eq. 6) to the gradient enables the learning to focus on hard positives and negatives. The gradient of the supervised loss with respect to $\boldsymbol{w}_i$ is related to that with respect to $\boldsymbol{z}_i$ via the chain rule:

$$\frac{\partial \mathcal{L}_i^{sup}(\boldsymbol{z}_i)}{\partial \boldsymbol{w}_i} = \frac{\partial \boldsymbol{z}_i}{\partial \boldsymbol{w}_i} \frac{\partial \mathcal{L}_i^{sup}(\boldsymbol{z}_i)}{\partial \boldsymbol{z}_i} \tag{10}$$

where:

$$\begin{aligned}
\frac{\partial \boldsymbol{z}_i}{\partial \boldsymbol{w}_i} &= \frac{\partial}{\partial \boldsymbol{w}_i}\left(\frac{\boldsymbol{w}_i}{\|\boldsymbol{w}_i\|}\right) \\
&= \frac{1}{\|\boldsymbol{w}_i\|}\mathbf{I} - \boldsymbol{w}_i\left(\frac{\partial\left(1/\|\boldsymbol{w}_i\|\right)}{\partial \boldsymbol{w}_i}\right)^T \\
&= \frac{1}{\|\boldsymbol{w}_i\|}\left(\mathbf{I} - \frac{\boldsymbol{w}_i \boldsymbol{w}_i^T}{\|\boldsymbol{w}_i\|^2}\right) \\
&= \frac{1}{\|\boldsymbol{w}_i\|}\left(\mathbf{I} - \boldsymbol{z}_i \boldsymbol{z}_i^T\right) \tag{11}
\end{aligned}$$

Combining Eqs. 6 and 11 thus gives:

$$\begin{aligned}
\frac{\partial \mathcal{L}_i^{sup}}{\partial \boldsymbol{w}_i} &= \frac{1}{\tau\|\boldsymbol{w}_i\|}\left(\mathbf{I} - \boldsymbol{z}_i \boldsymbol{z}_i^T\right)\left\{\sum_{p\in P(i)} \boldsymbol{z}_p(P_{ip} - X_{ip}) + \sum_{n\in N(i)} \boldsymbol{z}_n P_{in}\right\} \\
&= \frac{1}{\tau\|\boldsymbol{w}_i\|}\left\{\sum_{p\in P(i)} (\boldsymbol{z}_p - (\boldsymbol{z}_i \cdot \boldsymbol{z}_p)\boldsymbol{z}_i)(P_{ip} - X_{ip}) + \sum_{n\in N(i)} (\boldsymbol{z}_n - (\boldsymbol{z}_i \cdot \boldsymbol{z}_n)\boldsymbol{z}_i)P_{in}\right\} \\
&= \frac{\partial \mathcal{L}_i^{sup}}{\partial \boldsymbol{w}_i}\bigg|_{\text{P(i)}} + \frac{\partial \mathcal{L}_i^{sup}}{\partial \boldsymbol{w}_i}\bigg|_{\text{N(i)}} \tag{12}
\end{aligned}$$

where:

$$\frac{\partial \mathcal{L}_i^{sup}}{\partial \boldsymbol{w}_i}\bigg|_{\text{P(i)}} = \frac{1}{\tau\|\boldsymbol{w}_i\|}\sum_{p\in P(i)} (\boldsymbol{z}_p - (\boldsymbol{z}_i \cdot \boldsymbol{z}_p)\boldsymbol{z}_i)(P_{ip} - X_{ip}) \tag{13}$$

$$\frac{\partial \mathcal{L}_i^{sup}}{\partial \boldsymbol{w}_i}\bigg|_{\text{N(i)}} = \frac{1}{\tau\|\boldsymbol{w}_i\|}\sum_{n\in N(i)} (\boldsymbol{z}_n - (\boldsymbol{z}_i \cdot \boldsymbol{z}_n)\boldsymbol{z}_i)P_{in} \tag{14}$$

We now show that easy positives and negatives have small gradient contributions while hard positives and negatives have large ones. For an easy positive (i.e., one against which contrasting the anchor only *weakly* benefits the encoder), $\boldsymbol{z}_i \cdot \boldsymbol{z}_p \approx 1$. Thus (see Eq. 13):

$$\|(\boldsymbol{z}_p - (\boldsymbol{z}_i \cdot \boldsymbol{z}_p)\boldsymbol{z}_i\| = \sqrt{1 - (\boldsymbol{z}_i \cdot \boldsymbol{z}_p)^2} \approx 0 \tag{15}$$

However, for a hard positive (i.e., one against which contrasting the anchor *greatly* benefits the encoder), $\boldsymbol{z}_i \cdot \boldsymbol{z}_p \approx 0$, so:

$$\|(\boldsymbol{z}_p - (\boldsymbol{z}_i \cdot \boldsymbol{z}_p)\boldsymbol{z}_i\| = \sqrt{1 - (\boldsymbol{z}_i \cdot \boldsymbol{z}_p)^2} \approx 1 \tag{16}$$

Thus, for the gradient of $\mathcal{L}_{in}^{sup}$ (where $X_{ip} = X_{ip}^{in}$):

$$
\begin{aligned}
&\|(\boldsymbol{z}_p - (\boldsymbol{z}_i \cdot \boldsymbol{z}_p)\boldsymbol{z}_i\| \, |P_{ip} - X_{ip}^{in}| \\
&\approx |P_{ip} - X_{ip}^{in}| \\
&= \left| \frac{1}{\sum\limits_{p' \in P(i)} \exp\left(\boldsymbol{z}_i \cdot \boldsymbol{z}_{p'}/\tau\right) + \sum\limits_{n \in N(i)} \exp\left(\boldsymbol{z}_i \cdot \boldsymbol{z}_n/\tau\right)} - \frac{1}{\sum\limits_{p' \in P(i)} \exp\left(\boldsymbol{z}_i \cdot \boldsymbol{z}_{p'}/\tau\right)} \right| \\
&\propto \sum\limits_{n \in N(i)} \exp\left(\boldsymbol{z}_i \cdot \boldsymbol{z}_n/\tau\right)
\end{aligned}
\tag{17}
$$

For the gradient of $\mathcal{L}_{out}^{sup}$ (where $X_{ip} = X_{ip}^{out}$)

$$
\begin{aligned}
&\|(\boldsymbol{z}_p - (\boldsymbol{z}_i \cdot \boldsymbol{z}_p)\boldsymbol{z}_i\| \, |P_{ip} - X_{ip}^{out}| \\
&\approx |P_{ip} - X_{ip}^{out}| \\
&= \left| \frac{1}{\sum\limits_{a \in A(i)} \exp\left(\boldsymbol{z}_i \cdot \boldsymbol{z}_a/\tau\right)} - \frac{1}{|P(i)|} \right| \\
&= \left| \frac{1}{\sum\limits_{p' \in P(i)} \exp\left(\boldsymbol{z}_i \cdot \boldsymbol{z}_{p'}/\tau\right) + \sum\limits_{n \in N(i)} \exp\left(\boldsymbol{z}_i \cdot \boldsymbol{z}_n/\tau\right)} - \frac{1}{|P(i)|} \right| \\
&\propto \sum\limits_{n \in N(i)} \exp\left(\boldsymbol{z}_i \cdot \boldsymbol{z}_n/\tau\right) + \sum\limits_{p' \in P(i)} \exp\left(\boldsymbol{z}_i \cdot \boldsymbol{z}_{p'}/\tau\right) - |P(i)|
\end{aligned}
\tag{18}
$$

where $\sum_{n \in N(i)} \exp(\boldsymbol{z}_i \cdot \boldsymbol{z}_n/\tau) \geq 0$ (assuming $\boldsymbol{z}_i \cdot \boldsymbol{z}_n \leq 0$) and $\sum_{p' \in P(i)} \exp(\boldsymbol{z}_i \cdot \boldsymbol{z}_{p'}/\tau) - |P(i)| \geq 0$ (assuming $\boldsymbol{z}_i \cdot \boldsymbol{z}_{p'} \geq 0$). We thus see that for either $\mathcal{L}_{out}^{sup}$ and $\mathcal{L}_{in}^{sup}$ the gradient response to a hard positive in any individual training step can be made larger by increasing the number of negatives. Additionally, for $\mathcal{L}_{out}^{sup}$, it can also be made larger by increasing the number of positives.

Thus, for weak positives (since $\boldsymbol{z}_i \cdot \boldsymbol{z}_p \approx 1$) the contribution to the gradient is small while for hard positives the contribution is large (since $\boldsymbol{z}_i \cdot \boldsymbol{z}_p \approx 0$). Similarly, analysing Eq. 14 for weak negatives ($\boldsymbol{z}_i \cdot \boldsymbol{z}_n \approx -1$) vs hard negatives ($\boldsymbol{z}_i \cdot \boldsymbol{z}_n \approx 0$) we conclude that the gradient contribution is large for hard negatives and small for weak negatives.

In addition to an increased number of positives/negatives helping in general, we also note that as we increase the batch size, we also increase the probability of choosing individual *hard* positives/negatives. Since hard positives/negatives lead to a larger gradient contribution, we see that a larger batch has multiple high impact effects to allow obtaining better performance, as we observe empirically in the main paper.

Additionally, it should be noted that the ability of contrastive losses to perform intrinsic hard positive/negative data mining comes about only if a normalization layer is added to the end of the projection network, thereby justifying the use of a normalization in the network. Ours is the first paper to show analytically this property of contrastive losses, even though normalization has been empirically found to be useful in self-supervised contrastive learning.

## 4 Triplet Loss Derivation from Contrastive Loss

In this section, we show that the triplet loss [11] is a special case of the contrastive loss when the number of positives and negatives are each one. Assuming the representation of the anchor ($i$) and the positive ($p$) are more aligned than that of the anchor and negative ($n$) (i.e., $\boldsymbol{z}_i \cdot \boldsymbol{z}_p \gg \boldsymbol{z}_i \cdot \boldsymbol{z}_n$),

we have:

$$\begin{aligned}
\mathcal{L}^{self} &= -\log \frac{\exp\left(\boldsymbol{z}_a \cdot \boldsymbol{z}_p / \tau\right)}{\exp\left(\boldsymbol{z}_a \cdot \boldsymbol{z}_p / \tau\right) + \exp\left(\boldsymbol{z}_a \cdot \boldsymbol{z}_n / \tau\right)} \\
&= \log\left(1 + \exp\left(\left(\boldsymbol{z}_a \cdot \boldsymbol{z}_n - \boldsymbol{z}_a \cdot \boldsymbol{z}_p\right)/\tau\right)\right) \\
&\approx \exp\left(\left(\boldsymbol{z}_a \cdot \boldsymbol{z}_n - \boldsymbol{z}_a \cdot \boldsymbol{z}_p\right)/\tau\right) \quad \text{(Taylor expansion of log)} \\
&\approx 1 + \frac{1}{\tau} \cdot \left(\boldsymbol{z}_a \cdot \boldsymbol{z}_n - \boldsymbol{z}_a \cdot \boldsymbol{z}_p\right) \\
&= 1 - \frac{1}{2\tau} \cdot \left(\|\boldsymbol{z}_a - \boldsymbol{z}_n\|^2 - \|\boldsymbol{z}_a - \boldsymbol{z}_p\|^2\right) \\
&\propto \|\boldsymbol{z}_a - \boldsymbol{z}_p\|^2 - \|\boldsymbol{z}_a - \boldsymbol{z}_n\|^2 + 2\tau
\end{aligned}$$

which has the same form as a triplet loss with margin $\alpha = 2\tau$. This result is consistent with empirical results [1] which show that contrastive loss performs better in general than triplet loss on representation tasks. Additionally, whereas triplet loss in practice requires computationally expensive hard negative mining (e.g., [6]), the discussion in Sec. 3 shows that the gradients of the supervised contrastive loss naturally impose a measure of hard negative reinforcement during training. This comes at the cost of requiring large batch sizes to include many positives and negatives.

## 5 Supervised Contrastive Loss Hierarchy

The SupCon loss subsumes multiple other commonly used losses as special cases of itself. It is insightful to study which additional restrictions need to be imposed on it to change its form into that of each of these other losses.

For convenience, we reprint the form of the SupCon loss.

$$\mathcal{L}^{sup} = \sum_{i \in I} \frac{-1}{|P(i)|} \sum_{p \in P(i)} \log \frac{\exp\left(\boldsymbol{z}_i \cdot \boldsymbol{z}_p / \tau\right)}{\sum\limits_{a \in A(i)} \exp\left(\boldsymbol{z}_i \cdot \boldsymbol{z}_a / \tau\right)} \tag{19}$$

Here, $P(i)$ is the set of all positives in the multiviewed batch corresponding to the anchor $i$. For SupCon, positives can come from two disjoint categories:

- Views of the *same* sample image which generated the anchor image.
- Views of a sample image *different* from that which generated the anchor image but having the same label as that of the anchor.

The loss for self-supervised contrastive learning (Eq. 1 in the paper) is a special case of SupCon when $P(i)$ is restricted to contain only a view of the *same* source image as that of the anchor (i.e., the first category above). In this case, $P(i) = j(i)$, where $j(i)$ is the index of view, and Eq. 19 readily takes on the self-supervised contrastive loss form.

$$\mathcal{L}^{sup}\big|_{P(i)=j(i)} = \mathcal{L}^{self} = -\sum_{i \in I} \log \frac{\exp\left(\boldsymbol{z}_i \cdot \boldsymbol{z}_{j(i)} / \tau\right)}{\sum\limits_{a \in A(i)} \exp\left(\boldsymbol{z}_i \cdot \boldsymbol{z}_a / \tau\right)} \tag{20}$$

A second commonly referenced loss subsumed by SupCon is the N-Pairs loss [7]. This loss, while functionally similar to Eq. 20, differs from it by requiring that the positive be generated from a sample image *different* from that which generated the anchor but which has the same label as the anchor (i.e., the second category above). There is also no notion of temperature in the original N-Pairs loss, though it could be easily generalized to include it. Letting $k(i)$ denote the positive originating from a different sample image than that which generated the anchor $i$, the N-Pairs loss has the following form:

$$\mathcal{L}^{sup}\big|_{P(i)=k(i),\tau=1} = \mathcal{L}^{n\text{-}pairs} = -\sum_{i \in I} \log \frac{\exp\left(\boldsymbol{z}_i \cdot \boldsymbol{z}_{k(i)}\right)}{\sum\limits_{a \in A(i)} \exp\left(\boldsymbol{z}_i \cdot \boldsymbol{z}_a\right)} \tag{21}$$

It is interesting to see how these constraints affect performance. For a batch size of 6144, a ResNet-50 encoder trained on ImageNet with N-Pairs loss achieves an ImageNet Top-1 classification accuracy of 57.4% while an identical setup trained with the SupCon loss achieves 78.7%.

Finally, as discussed in Sec. 4, triplet loss is a special case of the SupCon loss (as well as that of the self-supervised and N-Pairs losses) when the number of positives and negatives are restricted to both be one.

## 6 Effect of Temperature in Loss Function

Similar to previous work [1, 8], we find that the temperature $\tau$ used in the loss function has an important role to play in supervised contrastive learning and that the model trained with the optimal temperature can improve performance by nearly 3%. Two competing effects that changing the temperature has on training the model are:

1. **Smoothness:** The distances in the representation space used for training the model have gradients with smaller norm ($||\nabla \mathcal{L}|| \propto \frac{1}{\tau}$); see Section 2. Smaller magnitude gradients make the optimization problem simpler by allowing for larger learning rates. In Section 3.3 of the paper, it is shown that in the case of a single positive and negative, the contrastive loss is equivalent to a triplet loss with margin $\propto \tau$. Therefore, in these cases, a larger temperature makes the optimization easier, and classes more separated.

2. **Hard positives/negatives:** On the other hand, as shown in Sec 3, the supervised contrastive loss has structure that cause hard positives/negatives to improve performance. Additionally, hard negatives have been shown to improve classification accuracy when models are trained with the triplet loss [6]. Low temperatures are equivalent to optimizing for hard positives/negatives: for a given batch of samples and a specific anchor, lowering the temperature relatively increases the value of $P_{ik}$ (see Eq. 4) for samples which have larger inner product with the anchor, and reduces it for samples which have smaller inner product.

We found empirically that a temperature of 0.1 was optimal for top-1 accuracy on ResNet-50; results on various temperatures are shown in Fig. 4 of the main paper. We use the same temperature for all experiments on ResNet-200.

## 7 Effect of Number of Positives

| 1 [1] | 3 | 5 | 7 | 9 | No cap (13) |
|---|---|---|---|---|---|
| 69.3 | 76.6 | 78.0 | 78.4 | 78.3 | 78.5 |

Table 1: Comparison of Top-1 accuracy variability as a function of the maximum number of positives $|P(i)|$ varies from 1 to no cap . Adding more positives benefits the final Top-1 accuracy. Note that with 1 positive, this is equivalent to the self-supervised approach of [1] where the positive is an augmented version of the *same sample*.

We run ablations to test the effect of the number of positives. Specifically, we take at most $k$ positives for each sample, and also remove them from the denominator of the loss function so they are not considered as a negative. We train with a batch size of 6144, so without this capping there are 13 positives in expectation(6 positives, each with 2 augmentatioins, plus other augmentation of anchor image). We train for 350 epochs. Table 1 shows the steady benefit of adding more positives for a ResNet-50 model trained on ImageNet with supervised contrastive loss. Note that for each anchor, the number of positives always contains one positive which is the same sample but with a different data augmentation; and the remainder of the positives are different samples from the same class. Under this definition, self-supervised learning is considered as having 1 positive.

## 8 Robustness

Along with measuring the mean Corruption Error (mCE) and mean relative Corruption Error [4] on the ImageNet-C dataset (see paper, Section 4.2 and Figure 3), we also measure the Expected Calibration Error and the mean accuracy of our models on different corruption severity levels. Table 2 demonstrates how performance and calibration degrades as the data shifts farther from the training distribution and becomes harder to classify. Figure 2 shows how the calibration error of the model

Figure 2: Expected Calibration Error and mean top-1 accuracy at different corruption severities on ImageNet-C, on the ResNet-50 architecture (top) and ResNet-200 architecture (bottom). The contrastive loss maintains a higher accuracy over the range of corruption severities, and does not suffer from increasing calibration error, unlike the cross entropy loss.

increases as the level of corruption severity increases as measured by performance on ImageNet-C [4].

| Model | | Test | 1 | 2 | 3 | 4 | 5 |
|---|---|---|---|---|---|---|---|
| Loss | Architecture | ECE | | | | | |
| Cross Entropy | ResNet-50 | 0.039 | 0.033 | 0.032 | 0.047 | 0.072 | 0.098 |
| | ResNet-200 | 0.045 | 0.048 | 0.036 | 0.040 | 0.042 | 0.052 |
| Supervised Contrastive | ResNet-50 | 0.024 | 0.026 | 0.034 | 0.048 | 0.071 | 0.100 |
| | ResNet-200 | 0.041 | 0.047 | 0.061 | 0.071 | 0.086 | 0.103 |
| | | Top-1 Accuracy | | | | | |
| Cross Entropy | ResNet-50 | 78.24 | 65.06 | 54.96 | 47.64 | 35.93 | 25.38 |
| | ResNet-200 | 80.81 | 72.89 | 65.28 | 60.55 | 52.00 | 43.11 |
| Supervised Contrastive | ResNet-50 | 78.81 | 65.39 | 55.55 | 48.64 | 37.27 | 26.92 |
| | ResNet-200 | 81.38 | 73.29 | 66.16 | 61.80 | 54.01 | 45.71 |

Table 2: **Top**: Average Expected Calibration Error (ECE) over all the corruptions in ImageNet-C [4] for a given level of severity (lower is better); **Bottom**: Average Top-1 Accuracy over all the corruptions for a given level of severity (higher is better).

## 9   Two stage training on Cross Entropy

To ablate the effect of representation learning and have a two stage evaluation process we also compared against using models trained with cross-entropy loss for representation learning. We do this by first training the model with cross entropy and then re-initializing the final layer of the network randomly. In this second stage of training we again train with cross entropy but keep the weights of the network fixed. Table 3 shows that the representations learnt by cross-entropy for a ResNet-50 network are not robust and just the re-initialization of the last layer leads to large drop in accuracy and a mixed result on robustness compared to a single-stage cross-entropy training. Hence both methods of training cross-entropy are inferior to supervised contrastive loss.

## 10   Training Details

In this section we present results for various ablation experiments, disentangling the effects of (a) Optimizer and (b) Data Augmentation on downstream performance.

| | Accuracy | mCE | rel. mCE |
|---|---|---|---|
| Supervised Contrastive | **78.7** | **67.2** | 94.6 |
| Cross Entropy (1 stage) | 77.1 | 68.4 | 103.7 |
| Cross Entropy (2 stage) | 73.7 | 73.3 | **92.9** |

Table 3: Comparison between representations learnt using Supervised Contrastive and representations learnt using Cross Entropy loss with either 1 stage of training or 2 stages (representation learning followed by linear classifier).

## 10.1 Optimizer

We experiment with various optimizers for the contrastive learning and training the linear classifier in various combinations. We present our results in Table 4. The LARS optimizer [13] gives us the best results to train the embedding network, confirming what has been reported by previous work [1]. With LARS we use a cosine learning rate decay. On the other hand we find that the RMSProp optimizer [10] works best for training the linear classifier. For RMSProp we use an exponential decay for the learning rate.

| Contrastive Optimizer | Linear Optimizer | Top-1 Accuracy |
|---|---|---|
| LARS | LARS | 78.2 |
| LARS | RMSProp | 78.7 |
| LARS | Momentum | 77.6 |
| RMSProp | LARS | 77.4 |
| RMSProp | RMSProp | 77.8 |
| RMSProp | Momentum | 76.9 |
| Momentum | LARS | 77.7 |
| Momentum | RMSProp | 76.1 |
| Momentum | Momentum | 77.7 |

Table 4: Results of training the ResNet-50 architecture with AutoAugment data augmentation policy for 350 epochs and then training the linear classifier for another 350 epochs. Learning rates were optimized for every optimizer while all other hyper-parameters were kept the same.

## 10.2 Data Augmentation

We experiment with the following data augmentations:

- **AutoAugment**: [2] A two stage augmentation policy which is trained with reinforcement learning for Top-1 Accuracy on ImageNet.

- **RandAugment**: [3] A two stage augmentation policy that uses a random parameter in place of parameters tuned by AutoAugment. This parameter needs to be tuned and hence reduces the search space, while giving better results than AutoAugment.

- **SimAugment**: [1] An augmentation policy which applies random flips, rotations, color jitters followed by Gaussian blur. We also add an additional step where we warping the image before the Gaussian blur, which gives a further boost in performance.

- **Stacked RandAugment**: [9] An augmentation policy which is based on RandAugment [3] and SimAugment [1]. The strategy involves an additional RandAugment step before doing the color jitter as done in SimAugment. This leads to a more diverse set of images created by the augmentation and hence more robust training which generalizes better.

and found that AutoAugment [5] gave us the highest Top-1 accuracy on ResNet-50 for both the cross entropy loss and supervised contrastive loss. On the other hand Stacked RandAugment [9] gives us highest Top-1 accuracy on ResNet-200 for both the cross entropy loss and supervised contrastive Loss. We conjecture this is happens because Stacked RandAugment is a stronger augmentation strategy and hence needs a larger model capacity to generalize well.

We also note that AutoAugment is faster at runtime than other augmentation schemes such as RandAugment [3], SimAugment [1] or StackedRandAugment [9] and hence models trained with AutoAugment take lesser time to train. We leave experimenting with MixUp [15] or CutMix [14] as future work.

| Contrastive Augmentation | Linear classifier Augmentation | Accuracy |
|---|---|---|
| AutoAugment | AutoAugment | 78.6 |
| AutoAugment | RandAugment | 78.1 |
| AutoAugment | SimAugment | 75.4 |
| AutoAugment | Stacked RandAugment | 77.4 |
| SimAugment | AutoAugment | 76.1 |
| SimAugment | RandAugment | 75.9 |
| SimAugment | SimAugment | 77.9 |
| SimAugment | Stacked RandAugment | 76.4 |
| RandAugment | AutoAugment | 78.3 |
| RandAugment | RandAugment | 78.4 |
| RandAugment | SimAugment | 76.3 |
| RandAugment | Stacked RandAugment | 75.8 |
| Stacked RandAugment | AutoAugment | 78.1 |
| Stacked RandAugment | RandAugment | 78.2 |
| Stacked RandAugment | SimAugment | 77.9 |
| Stacked RandAugment | Stacked RandAugment | 75.9 |

Table 5: Combinations of different data augmentations for ResNet-50 trained with optimal set of hyperparameters and optimizers. We observe that stacked RandAugment does consistently worse for all configurations due to lower capacity of ResNet-50 models. We also observe that for other augmentations that we get the best performance by using the same augmentations in both stages of training.

Further we experiment with varying levels of augmentation magnitude for RandAugment since that has shown to affect performance when training models with cross entropy loss [3]. Fig. 3 shows that supervised contrastive methods consistently outperform cross entropy training independent of augmentation magnitude.

Figure 3: Top-1 Accuracy vs RandAugment magnitude for ResNet-50 (left) and ResNet-200 (right). We see that supervised contrastive methods consistently outperform cross entropy for varying strengths of augmentation.

## Footnotes

* Work done during Google AI Residency. Equal contribution.

† Work done while at Google Research.

‡ Corresponding author: sarna@google.com

[4]Note that when the normalization is combined with an inner product (as we do here), this is equivalent to cosine similarity. Some contrastive learning approaches [1] use a cosine similarity explicitly in their loss formulation. We decouple the normalization here to highlight the benefits it provides.