[Reviews · NeurIPS 2020]

Review 1

Summary and Contributions: The paper proposes using a contrastive loss for supervised image classification, by taking samples from the same class as positives. The paper shows that the new loss performs better than standard cross-entropy loss on standard image classification tasks.

Strengths: The idea of incorporating contrastive loss in supervised learning is good. The paper thoroughly discussed different ways to achieve this and provide in-depth analysis. Empirical evaluation shows that the proposed loss function is beneficial in practice. The paper also provides additional analysis on hyperparameter stability and transfer learning performance, both help understand the method better.

Weaknesses: I have some concerns regarding the training cost. Since the proposed method uses a "multiviewed batch" which is 2x the standard batch used by cross-entropy loss, its training cost is 2x the baseline. Using more compute cost (together with hyperparameter tuning) could be beneficial for training the baselines as well. The results would be more convincing if the comparison is performed under similar compute cost, e.g. using half as many epochs as the baseline. Though the paper claims state-of-the-art performance, it is largely due to a well-tuned baseline setting with autoaugment, large number (700) of epochs, cosine LR decay (only mentioned in supplementary, not clear if used in the baseline), etc. The improvement made from the proposed method is about 0.5% according to Table. 3, which is good but not very impressive, especially given the sensitivity to LARS optimizer shown in Table. 4 of the supplementary. The 700 epochs of training with 6k batch size (12k "multiviewed" images) is also not very practical to most members of the community. The contribution could be more practical (and consequently have more significance to the community) if similar improvements can be shown under friendly compute budget and a more regular batch size - which may potentially also remove the requirement of LARS optimizer. ===== update: author's response did well addressing my concern on the method's cost

Correctness: The method is formulated correctly and the experiments can support the claim.

Clarity: The paper has very good clarity.

Relation to Prior Work: Yes

Reproducibility: Yes

Additional Feedback: Given that supervision is used in the first stage of contrastive training to discriminate between classes, it would be good to also see the classification performance of the first stage, by, e.g., a kNN classifier.


Review 2

Summary and Contributions: The paper extends contrastive loss to the fully-supervised setting. It compares the new contrastive loss with the cross-entropy loss under several image classification datasets. It also evaluates the learned representation for fine-tuning on 12 natural image datasets.

Strengths: 1. The idea is simple and technically sound. 2. The experiments look solid. 3. Extensive ablation studies compare Supervised Contrastive and Cross-entropy with different augmentation, optimizer, learning rate, batch size, data size, and temperature.

Weaknesses: 1. What is the computation cost compared to cross-entropy loss? 2. According to Table.4, it seems that cross-entropy loss still a bit better than Supervised contrastive. Also, the cross-entropy loss is simpler than Supervised contrastive because Supervised contrastive needs to learn a linear classifier on top of the feature. What circumstance do you think Supervised contrastive can replace cross-entropy loss? 3. Authors have provided classification results for transfer learning. However, as ImageNet pretrain is widely used in a lot of application and contrastvie loss have shown superior transfer performance on some task, I wonder how does Supervised Contrastive loss behave as a pre-training model for other vision tasks.(e.g. Object Detection) 4. Is cross-entropy loss and supervised contrastive loss learn similar pattern? If not, can we combine them to achieve better performance? 5. I understand that surpass cross-entropy loss is hard and the value of this paper. But my major concern is that supervised contrastive loss have very close performance to cross-entropy loss. It would be good if authors can provide more promising results. --------_After rebuttal----------- The authors address my concern. I would keep my rating as accept.

Correctness: Yes.

Clarity: Yes.

Relation to Prior Work: Yes/

Reproducibility: Yes

Additional Feedback:


Review 3

Summary and Contributions: The paper introduces a variation of self-supervised learning when labeled information is also available. The main idea is to regard data points of the same label as additional positive pairs, on top of the existing pairs that are just doing instance level discrimination. Empirically, it is shown to outperform the normal cross-entropy loss that are prevailing current approaches consistently, and shows better robustness.

Strengths: + The paper shows consistent and robust gains over the baseline cross-entropy loss. + I like the style of the paper where the story is interleaved with empirical justifications, e.g. whether to use L_in or L_out, and Table 1 gives a convincing answer. + Overall the paper is well written and well organized. It is easy to follow the main idea in the main paper, and some side concerns (e.g. connection to triplet loss) are addressed in the supplementary material. + The code will be released for open research.

Weaknesses: - It seems one major downside of this contrastive learning framework is that it needs to be trained much longer than supervised baselines (700 ep vs 200 ep in general), this is costly and environmental unfriendly. I hope this concern can be addressed in the rebuttal. - Some of the table designs in the paper is a bit confusing and even misleading. For example Table 3, it would be most fair to compare methods *grouped by* architecture and augmentations, e.g. the ResNet 50 model (w/ auto augment) should have the rows CE and SC together, so that it is easier to compare between the two. Otherwise it is not immediately clear why the third last row should be bold, and the 2nd last row not. For Table 4, I also think it is important to high-light that SimCLR (a self-supervised method) can achieve really good result (actually on a majority of the datasets it is the best method and should be bolded but the current table highlights the comparison between supervised methods only) - I also believe it would be great to fix the optimizer for a more fair comparison -- or at least show the same SGD setting of contrastive learning and LARS setting for supervised CE. Otherwise the claims are not backed up to me.

Correctness: Most claims in the paper are supported by empirical results or math derivations. Although I haven't checked carefully, I believe they are correct. On the other hand, the empirical set of experiments are well designed and mostly sound reasonable to me.

Clarity: Duplicate question, see answer above -- yes it is well written.

Relation to Prior Work: The paper addresses the relation to previous contributions well, except that there can be quite a few works that explore different losses for supervised learning as well and it is not mentioned. For example, Cui, Yin, et al. "Class-balanced loss based on effective number of samples." Proceedings of the IEEE Conference on Computer Vision and Pattern Recognition. 2019. This work explored modified focal loss and show it can also be used for ImageNet training. I believe the paper would benefit from discussions to such prior works.

Reproducibility: Yes

Additional Feedback:


Review 4

Summary and Contributions: This paper proposed supervised contrastive learning for image classification task, achieving state-of-the-art performance. The proposed loss function can form multiple positive and negative pairs for each anchor in a minibatch, leading to more effective training. Detailed analysis for the proposed method is presented. The experiments show the clear improvements on ImageNet classification.

Strengths: The proposed method is conceptually simple and intuitively effective. By forming more positive pairs can make better use of the data. Without data-mining, the learning process is much more efficient. The experiments are comprehensive. The improvements are validated on multiple datasets. The method is presented clearly, with enough details and analysis.

Weaknesses: The motivation is unclear for me. In introduction and related work sections, self-supervised learning is mentioned multiple times. This paper aims to extend the self-supervised contrastive approach to fully-supervised setting. However, supervised contrastive learning is originally proposed in [1] and has been applied to many applications [2][3][4], etcs. I didn't see the motivation for this extension (self-supervised to fully supervised). If it is an extension from contrastive loss, it should be included in experimental comparison as well. From my perspective, the proposed loss function is for feature learning, not classification, since a linear classifier has to be learned in the second stage. In this case, I would consider it as a deep metric learning method, so it is expected to compare with triplet loss, N-pairs loss, [5] and other advanced metric learning methods. [1] Dimensionality Reduction by Learning an Invariant Mapping [2] Learning a similarity metric discriminatively, with application to face verification [3] Deep learning face representation by joint identification-verification [4] Discriminative Feature Representation for Person Re-identification by Batch-contrastive Loss [5] Deep Metric Learning via Lifted Structured Feature Embedding

Correctness: yes

Clarity: yes

Relation to Prior Work: yes

Reproducibility: Yes

Additional Feedback: The rebuttal has addressed my concerns. So I would keep my decision "accept" unchanged.

[Author Response · NeurIPS 2020]

We thank the reviewers for their positive comments and detailed feedback. Specific concerns are addressed below.

**R4: "Motivation unclear; compare to earlier metric learning work":** In our related work (lines 97-109), we outline the differences between modern contrastive learning methods (including ours) and the papers listed by R4. In summary: our approach can be seen as a generalization of the prior approaches by using multiple positives and negatives per anchor; the use of a temperature parameter which plays the role of a margin; and the use of multiple views of the data. Each of these components contributes to the performance of the supervised contrastive loss, as shown in our ablation experiments. It is well known that distance metric learning approaches such as triplet loss and the loss proposed in [4] have slow convergence and require proper tuning of hard negatives (e.g. see [1], Section 3). Additionally, some of these losses seem to require very small batch sizes to avoid falling into local minima (e.g., [3] cites a batch size of 16 for this exact reason). Our approach overcomes these issues as shown by our application to large scale problems such as ImageNet. In Section 3.2.3, we show analytically the reduction to the triplet loss in Section 3.2.3. We will further clarify these points in the final version.

**R4: "Comparison to metric learning methods":** We tested N-pairs [2] in our framework with a batch size of 6144. N-pairs achieves only 57.4% Top-1 on ImageNet (compare to our loss that achieves 78.7%). We believe this is due to multiple factors missing from N-pairs loss compared to supervised contrastive: the use of multiple views; lower temperature (e.g., see [5], Table 5) and many more positives. We show some results of the impact of the number of positives per anchor in the Appendix (Sec. 6), and the N-pairs result is inline with them. We also note that the original N-pairs paper [2] has already shown the outperformance of N-pairs loss to triplet loss. We will add this experiment and relevant ablations to the paper.

**R1/R2/R3: "Concerns over compute cost":** We agree with the reviewers that reducing the expense of contrastive methods in general would be desirable. To this end, we experimented with memory based alternatives [1]. On ImageNet, with a memory size of 8192 (requiring only the storage of 128-dimensional vectors), a batch size of 256, and SGD optimizer, running on 8 Nvidia V100 GPUs, the supervised contrastive loss is able to achieve 79% top-1 accuracy on ResNet-50 architecture. This is in fact slightly better than the 78.7% accuracy with 6144 batch size (and no memory); and with significantly reduced compute and memory footprint. We will add this experiment to the paper. Finally, the drop in performance by training for fewer epochs (e.g. 350 vs 700 epochs) is very small (difference of 0.1%). We will add a sweep over training epochs to the paper.

**R3: "Experiment reporting is confusing":** We agree with R3 and will adjust the table reporting to be more clear and will highlight further SimCLR's performance for transfer learning.

**R1/R3: "Improved baselines for cross-entropy; additional experiments":** R3 requested that we show comparisons of cross-entropy and supervised contrastive losses using the same optimizers. In Table 4 of the Supplementary we include a sweep over optimizers used for supervised contrastive. In Table 1, we provide an equivalent sweep over optimizers used for cross-entropy using the same network, augmentation strategy, and number of train epochs; cross-entropy performs best with the momentum optimizer (the results show no improvement over those reported in the paper). We additionally ran experiments to see if the larger effective batch size used for supervised contrastive could explain its superior performance, as R1 suggested. We trained with cross-entropy loss using a batch size of 12,288, but this only achieved 77.5% Top-1 accuracy. We also tried training for twice as many epochs (1400) with the original batch size of 6,144, but this was even lower, at 77.0%. We will add these results to the final version. Please note in Figure 1 and Table 3 of the paper, baseline numbers are taken directly from previous papers.

**R2: "Table 4 cross-entropy outperformed Supervised Contrastive; consider a joint loss":** Note that Table 4 is about transfer learning where we see that SimCLR (unsupervised contrastive learning) actually outperforms both cross-entropy and supervised contrastive slightly. We agree with R2 that a joint cross-entropy and supervised contrastive loss could probably combine the benefits of both and certainly make it simpler to use the supervised contrastive loss in practice. We have experimented with this and seen strong results, although none that so far outperform the two stage approach. We will add a discussion of this in the final paper, but we feel that it's important for the sake of future research to isolate the impact of the supervised contrastive loss, so we chose to make the two-stage approach the focus of the paper.

| Optimizer | Top-1 Accuracy |
|-----------|----------------|
| LARS | 76.2% |
| RMSProp | 76.2% |
| Momentum | 78.2% |

Table 1: Cross-entropy (ResNet-50) with different optimizers.

# References

[1] Momentum Contrast for Unsupervised Visual Representation Learning, He et. al., 2019.

[2] Improved Deep Metric Learning with Multi-class N-pair Loss Objective, K. Sohn, 2016.

[3] Discriminative Feature Representation for Person Re-identification by Batch-contrastive Loss, Zhang et. al., 2018.

[4] Dimensionality reduction by learning an invariant mapping, Hadsell et. al., 2006.

[5] A Simple Framework for Contrastive Learning of Visual Representations, Chen et. al., 2020.


[Meta-Review · NeurIPS 2020]

Four knowledgeable reviewers support acceptance for the contributions. Reviewers find the paper well-written and appreciate the strong empirical results by proposing a loss function that can form multiple positive and negative pairs for each anchor in a minibatch, leading to more effective training. Therefore, I also recommend acceptance. However, please consider revising your paper to address all the concerns and comments from the reviewers.